# Caregiver Burden and Quality of Life in Late Stage Parkinson’s Disease

**DOI:** 10.3390/brainsci12010111

**Published:** 2022-01-14

**Authors:** Kristina Rosqvist, Anette Schrag, Per Odin

**Affiliations:** 1Restorative Parkinson Unit, Division of Neurology, Department of Clinical Sciences Lund, Faculty of Medicine, Lund University, Wigerthuset, Remissgatan 4, 221 85 Lund, Sweden; per.odin@med.lu.se; 2Department of Neurology, Skåne University Hospital, 222 42 Lund, Sweden; 3Queen Square Institute of Neurology, University College London, London WC1N 3BG, UK; a.schrag@ucl.ac.uk

**Keywords:** Parkinson’s disease, late stage, informal caregiver, caregiver burden

## Abstract

Parkinson’s disease (PD) is a chronic, progressive, neurodegenerative disease involving both motor and non-motor symptoms (NMS). In the late stage of the disease, Hoehn and Yahr (HY) stages IV-V, the symptomatology is often severe and patients become increasingly dependent on help in their daily life, resulting in an increased burden for the informal caregivers. To assess the implications of the caregiver burden, caregiver quality of life (QoL) was assessed in 74 informal caregivers to patients in late stage PD, by the Alzheimer’s Patient Partners Life Impact Questionnaire (APPLIQue), which has been found useful also in PD. The majority of caregivers were the spouse/partner. Individual items provided information on which aspects of caregiver burden were the most common, i.e., items: “feel guilty if not there” (71% affirmed), “situation wears me down” (65% affirmed) and “always on my mind” (61% affirmed). In simple linear regression analyses, female patient gender (*p* = 0.007), better cognition (*p* = 0.004), lower NMS burden (*p* = 0.012) and not being the partner (*p* = 0.022) were associated with better caregiver QoL. Multivariable linear regression analyses identified better cognition (*p* = 0.004) and female patient gender (*p* = 0.035) as independently associated with better informal caregiver QoL. Identifying and treating NMS as well as recognizing and alleviating caregiver burden seem essential to enhance QoL for both patients and caregivers in late stage PD.

## 1. Introduction

Parkinson’s disease (PD) is a chronic, progressive neurological disease, involving both motor and non-motor symptoms (NMS) [1,2]. In the late stage of the disease, i.e., Hoehn and Yahr (HY) stages IV and V [3], motor and NMS are pronounced and the patients become increasingly dependent on help from others in activities of daily living (ADL) [4,5]. Informal caregiver burden increases when the disease progresses and the patient becomes more dependent on the informal caregiver in everyday activities [6].

Informal caregivers make an essential contribution to the wellbeing and everyday functioning of patients in late stage PD [7]. As a large proportion (around 80%) of late stage PD patients live at home [8,9], it is of importance to acknowledge and support the needs not only of the late stage PD patients, but of their informal caregivers, who describe themselves as “the engine” which keeps everyday life running and make living at home possible in the late stage of the disease [10]. The high caregiver burden is likely also to affect caregivers’ quality of life (QoL).

There is a need to further investigate the caregiver burden and QoL in late stage PD, to add quantitative perspectives on informal caregiver burden and QoL to those that were previously explored qualitatively within the Swedish late stage cohort [10]. It is therefore of importance to investigate and analyse how informal caregivers in late stage PD rate their QoL, with regard to the burden of caregiving. The aim of this study was to describe and assess factors associated with informal caregiver QoL in late stage PD.

## 2. Materials and Methods

Late stage PD was defined as HY stages IV and V (score range 1–5, higher = worse) [3] while in the “on” medication state and/or having a substantial need of help with ADL, i.e., ≤50% on the Schwab and England ADL Scale (score range 0–100, higher = better) [11].

The participants were recruited in the southern region of Sweden through neurology departments and in a few cases through the municipality-based health care system. This cohort constitutes the Swedish part of the European multicenter Care of Late Stage Parkinsonism (CLaSP) project [4,12]. The current analyses were specific for the Swedish sub-study of the CLaSP project and add quantitative perspectives to already published qualitative data on caregiver burden and QoL in late stage PD [10] and previously published data on caregiver burden in the international sample [13].

Informal caregiver QoL was assessed with the Alzheimer’s Patient Partners Life Impact Questionnaire (APPLIQue, score range 0–25, higher = worse) [14,15], a development of the Alzheimer’s Carers’ QoL Inventory (ACQLI, score range 0–30, higher = worse), found useful also in other neurodegenerative diseases [16], as at the time of the study design in 2013 there were no specific PD caregiver QoL instruments. The patients’ clinical status was assessed with the following instruments. Cognition was assessed with the Mini-Mental State Examination (MMSE, score range 0–30, higher = better) [17]. Non-motor symptomatology was assessed with the NMS Scale (NMSS, score range 0–360, higher = worse), which consists of 30 items where each item is scored regarding frequency and severity 0–12 [18]. As it has been found that neuropsychiatric symptoms highly affect informal caregiver burden [13], the individual items hallucinations (item 13) and delusions (item 14) were investigated specifically and the score was presented as not present, 1–5 (mild), ≥6 moderate/severe (moderate ≥ 6/severe ≥ 8) and were dichotomized as 0 vs. ≥1. Depressive symptoms were assessed by the Geriatric Depression Scale (GDS-30; score range 0–30, higher = worse) [19]. Motor function was assessed with the motor part of the Unified PD Rating Scale [20], part III (UPDRS III, score range 0–108, higher = worse).

### Statistical Analyses

Descriptive and clinical data are presented as median with first and third quartiles (q1–q3) and frequencies and percentages, as appropriate. Associations were tested statistically with simple linear regression analyses. Independent variables with *p*-values < 0.3 from the simple linear regression analyses were simultaneously entered into a multivariable linear regression model to identify factors independently associated with informal caregiver QoL, where the dependent variable was the APPLIQue total score. A backward-stepping regression analysis was conducted where *p*-values were inspected and the highest *p*-value was manually removed from the model, which was repeated until the remaining independent variables in the model had *p*-values < 0.1. *P*-values < 0.05 were considered statistically significant. All analyses were performed using IMB SPSS version 26.0 (IBM Corporation, Armonk, NY, USA).

## 3. Results

### 3.1. Demographic and Clinical Data

The present study consisted of 74 informal caregivers from a cohort of 107 patients in late stage PD. The majority of them (59; 80%) were the spouse/partner, 10 (14%) were a daughter, four (5%) were a son and one was a sibling to the patient. The median (q1–q3) patient age was 78 (73–84) years and the median disease duration was 15 (11–19) years. The median (q1–q3) UPDRS III score was 40 (29–53); the median (q1–q3) NMSS score was 91 (55–128); the median (q1–q3) MMSE score was 22 (18–27) and 60 (58%) of the participants scored below the screening cutoff for cognitive impairment of ≤23 points. The median (q1–q3) GDS-30 score was 11 (8–16), where 62 (62%) of the participants were above the screening cutoff (≥10) for depression. 

Almost two thirds of the patients (*n* = 67; 63%) lived in ordinary housing and 40 (37%) in a nursing home, and 65 (61%) had a spouse/partner. The median (q1–q3) caregiver QoL (APPLIQue) score was 8 (3–12). APPLIQue scores for spousal/partner caregivers (*n* = 59) were considerably higher (i.e., rating worse QoL; median, q1–q3: 8; 4–14) compared to the group child/sibling (*n* = 15), where the informal caregiver did not live with the patient (median, q1–q3: 2; 0–8) (Table 1).

The scoring of the individual items of the APPLIQue questionnaire provided information on which aspects of caregiver burden were rated as the most common, i.e., items “feel guilty if not there” (71% affirmed), “situation wears me down” (65% affirmed) and “always on my mind” (61% affirmed) (Figure 1).

### 3.2. Simple and Multivariable Linear Regression Analyses

In the simple linear regression analyses, better cognition (*p* = 0.004), female patient gender (*p* = 0.007), lower NMS burden (*p* = 0.012) and not being the partner of the patient (*p* = 0.022) were significantly associated with better informal caregiver QoL (Table 2). The multivariable linear regression analyses identified better cognition (*p* = 0.004) and female patient gender (*p* = 0.035) as independently associated with better informal caregiver QoL. Higher patient age (*p* = 0.082) was furthermore associated with better informal caregiver QoL, although not statistically significant on the *p* = 0.05 level (Table 3).

## 4. Discussion

The current analyses of caregiver QoL in the Swedish sub-study of the CLaSP project [4,12] adds quantitative perspectives on informal caregiver burden and QoL in late stage PD to those that were previously explored qualitatively within the Swedish late stage cohort [10].

The present results reveal and confirm that the burden on informal caregivers in late stage PD is pronounced. Being the informal caregiver of a person suffering from a chronic, progressive, neurodegenerative disease such as PD greatly influences the life of the informal caregiver, where expressions such as “situation wears me down”, “always on my mind” and “feel guilty if not there” were affirmed by 60–70% of the informal caregivers.

The univariate results showed that female patient gender, better cognition, lower NMS burden and not being the partner of the patient were associated with better informal caregiver QoL. Motor severity in patients was however not a factor associated with caregiver QoL, at least not in this cohort who were all in advanced stages of the disease. When using the well-established cutoff at 23/24 of the MMSE in screening for cognitive impairment, there was a clear association with caregiver QoL, where better cognitive function showed the strongest association with better caregiver QoL. This is in line with previous studies, where cognitive impairment has been consistently associated with increased caregiver burden [7,21] as well as a major predictor of disability [4]. Cognitive impairment may entail deficits of attention, language, memory and executive functioning as well as behavioral and psychological manifestations [7]. 

Factors such as whether the patient had depressive symptoms or showed more pronounced motor symptomatology did not come through among the factors associated with caregiver QoL, which is in line with previous research on late stage PD [22]. Previous research also showed patients’ depression as only moderately associated with caregiver burden and weakly associated with informal caregiver health-related QoL [23]. However, when looking at the QoL score, there is a slight tendency that these factors may also influence caregiver QoL, as informal caregivers had slightly worse QoL when the patient had more pronounced motor symptoms as well as depressive symptoms compared to less of these symptoms. Moreover, mood/apathy is one of the domains of the NMSS, which total score was significantly associated with informal caregiver QoL. This is in line with previous research that show that NMS consistently have greater impact on informal caregiver QoL than motor symptoms [7,13]. Previous research moreover indicated that there is an increase in informal caregiver burden when the patient manifests many neuropsychiatric symptoms as well as other NMS [13].

We have previously shown in the analyses of the whole international late stage cohort that the strongest correlations with caregiver burden on the Zarit Caregiver Interview were found for impairments in the NMS domains attention/memory and mood/apathy [13]. Other NMS features such as sleep disturbances, impulse-control disorders, anxiety and depressive symptoms have all, in previous studies, shown negative effect on caregiver QoL [7,24]. Management of neuropsychiatric symptoms has been identified as important in order to reduce informal caregiver burden in PD [7,13,24]. In the present analyses special focus was placed on investigating associations with neuropsychiatric symptomatology and informal caregiver QoL but we did not find that this was associated with worse caregiver QoL. Since neuropsychiatric as well as other NMS are frequent and progressive in late stage PD [4,25,26], it is essential to identify and attempt to treat these symptoms in order to enhance the patients’ QoL [27], as patient QoL is frequently associated with informal caregiver burden [28]. Previous analyses suggest that current treatment in late stage PD to alleviate disabling symptoms is insufficient in many patients [4].

Caring for a male patient has been reported to be a significant predictor of greater caregiver burden, with female spouses experiencing greater caregiver burden than male spouses [6]. This was the case also for the international late-stage cohort, which the current sample is part of [13]. Others have found no difference between male and female caregiver burden [29]. The present study further investigated how informal caregivers in late stage PD rate their QoL, with regard to the burden of caregiving. 

While some studies indicate that female gender in caregivers may predict greater risk for increased informal caregiver burden [30], it may be important to differentiate informal caregiver burden from QoL. The caregiver QoL in this study was clearly reduced when the patient was male, which in most cases meant that the informal caregiver was a female. It is unclear whether this signifies that the informal caregiver burden is experienced more negatively in female than male caregivers, resulting in worse QoL, or whether there is a genetic or cultural predisposition for female caregivers in taking on a larger amount of informal caregiver work, resulting in a larger burden and worse QoL than a male informal caregiver. The previous qualitative interview study on the current sample indicated that some of the female patients expressed that their male spouse sometimes showed irritation and anger in their caregiving role [10]. Caregiver mood has been shown to be the most important factor influencing caregiver burden and caregivers’ perceived health. Moreover, the prevalence of anxiety and depression was shown to be higher in PD caregivers than in the general population [31].

There are likely international differences regarding caregiving. In a Spanish study, most caregivers were women [31], which may be due to the prevalence of slightly more male PD patients, even though women generally live longer. Traditional gender roles, where females generally have been the primary caregivers, are likely to change in modern society and hence also the informal caregiving constellation in the future.

Limitations in performing activities on their own and difficulty maintaining relationships with friends were described among informal caregivers in late stage PD [10]. In spite of the great impact on their own mood and life in general, informal caregivers in late stage PD clearly expressed that the joy and benefits of still being together, despite the burden of the disease, outweighed the strain and limitations of the informal caregiving [10]. Family functioning with balanced cohesion has been found to be associated with reduced caregiver burden in a study on caregiver burden in different neurological diseases [32]. 

In a previous study on the current sample, satisfaction with support was associated with informal caregivers’ own QoL. Moreover, satisfaction with support was associated with patient HY stage, showing that informal caregivers were more satisfied with their support when the patient was in a more severe disease stage, i.e., HY stage V compared to IV [33]. In addition, in this study patient age tended to be associated with caregiver QoL in the final multivariable model, indicating that as the patient gets older, QoL tends to be better for informal caregivers. A higher amount of home health and social care as well as possibly an increasing acceptance of the situation may be part of the explanation for this.

Despite the presence of professional care resources, in Sweden often highly prevalent in late stage PD [33], the overall responsibility of care often rests with the informal caregivers. From the recent qualitative interviews with patients and informal caregivers in late stage PD, it was clear that both the patients and the informal caregivers wished for the patient to remain living at home as compared to in a residential care facility [10]. However, both the late stage patients and their informal caregivers expressed an awareness that the patient would not have been able to remain living at home without the informal caregiver [10,34]. It is therefore of great importance to recognize and alleviate the informal caregiver burden in late stage PD, as failure to do so may lead to informal caregiver exhaustion and premature institutionalization of the patient [7]. This could be done through information on available support systems within the community, involving increased home health and social care for the patient and possibility of support and respite for the informal caregivers. Major concerns of informal caregivers have been described to regard the safety of the patient, in the form of constant vigilance and worry. It is essential to recognize the mental burden on spousal caregivers, to improve their QoL. Education for caregivers in PD has therefore been suggested to involve methods of coping with mental stressors [35], as another step in supporting the valuable care performed by the spouses and family members to patients in late stage PD.

### Strengths, Limitations and Future Perspectives

This is one of the first studies on caregiver QoL in late stage PD with detailed information on how informal caregivers experience their situation regarding caregiver burden and their own QoL. As no PD-specific caregiver QoL instrument was available, we used the APPLIQue, which has been validated in caregivers of patients with Alzheimer’s disease. Whilst it may not capture all aspects of QoL in caregivers of patients with PD, it provides information highly relevant to caregivers of patients with late stage PD. The authors of the instrument suggested that caregiver burden is quite similar across neurodegenerative diseases such as PD, Alzheimer’s and Huntington’s, since they are all to varying degree associated with mobility problems, cognitive impairment and behavioral change, which could imply similarities in impact on family caregivers [15]. The questionnaire has been recommended as a measure of caregiver QoL in intervention studies [14]. They furthermore suggested that the APPLIQue could be a starting point in developing a common caregiver assessment instrument across neurodegenerative diseases [15]. 

Future studies should continue to investigate and illuminate the situation of informal caregivers in chronic, progressive neurological diseases such as PD, as the increasing disease burden not only affects the patients but to a high degree also their informal caregivers. Better understanding of their situation has the potential to enhance QoL for both the patients and their caregivers, in supporting them to carry on at home in spite of a progressive disease. In addition to relieving the workload itself, future interventions to reduce caregiver burden may include mindfulness and psychological flexibility training to help PD caregivers better cope with their burdensome situation [36].

Assessing the impact of informal caregiving on the QoL of the informal spousal caregiver should be carried out through future longitudinal analyses and randomized controlled trials, in various national contexts, as the situation and possibilities for support for the informal caregivers in late stage PD may vary greatly between countries.

## 5. Conclusions

The present informal caregiver data showed that there is a great burden on the informal caregivers of patients in late stage PD, resulting in decreased QoL, increasingly so when the patient is cognitively affected, has a high NMS burden and the caregiver is female. Spousal caregivers are most affected. Identifying and treating NMS, particularly related to cognitive impairment, as well as recognizing and alleviating caregiver burden seem essential to enhance QoL for both patients and caregivers, as PD is not a disease that affects only the patients, though it is to a high degree also an informal caregiver disease.

## Figures and Tables

**Figure 1 brainsci-12-00111-f001:**
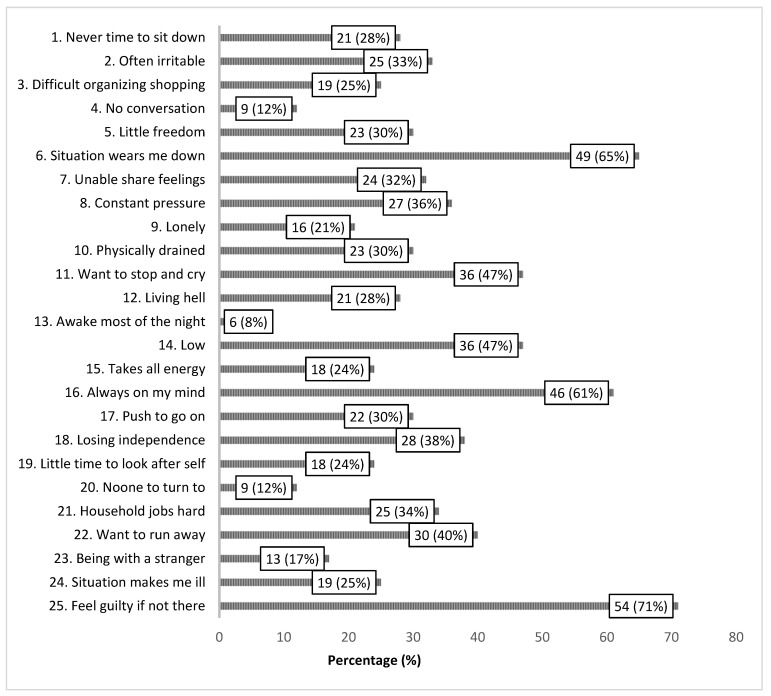
Self-reported quality of life of informal caregivers to patients in late stage Parkinson’s disease (APPLIQue 25 items; response alternatives yes/no), *n* = 74. Frequencies of informal caregivers responding yes (%).

**Table 1 brainsci-12-00111-t001:** Descriptive characteristics of patient data and in relation to informal caregiver self-reported QoL in late stage PD (APPLIQue), *n* = 74.

Variables	Patient Data	APPLIQue Score
	(*n* = 107)	(Median, q1–q3)
Age (year), median (q1–q3)	78 (73–84)	
PD duration, median (q1–q3)	15 (11–19)	
Gender		
Male	62 (58%)	8 (5–15)
Female	45 (42%)	3 (0–10)
Partner, *n* (%)	65 (61%)	
Dwelling place		
Home	67 (63%)	8 (3–15)
Nursing home	40 (37%)	7 (3–11)
Hoehn and Yahr stage		
IV	79 (74%)	7 (3–12)
V	28 (26%)	9 (4–16)
ADL independency (S&E), median (q1–q3)	40 (30–50)	
Home health care, *n* (%)		
No	18 (17%)	6 (3–12)
Yes	89 (83%)	8 (3–14)
Cognition (MMSE), median (q1–q3)	22 (18–27) *	
Cognitive impairment ≤ 23	60 (58%)	8 (6–14)
No cognitive impairment ≥ 24	43 (42%)	3 (0–9)
Depressive symptoms, GDS-30	11 (8–16) ***	
No depression (<10)	38 (38%)	6 (3–14)
Depression (≥10)	62 (62%)	8 (2–11)
Motor function (UPDRS III total score)	40 (29–53)	
Less symptoms (first and second quartiles)		7 (3–11)
More symptoms (third and fourth quartiles)		8 (3–14)
Non-motor symptomatology (NMSS total score)	91 (55–128) **	
Less symptoms (first and second quartiles)		6 (0–14)
More symptoms (third and fourth quartiles)		8 (5–12)
Hallucinations (NMSS item 13), median (q1–q3)	0 (0–4)	
Not present	58 (54%)	
Mild (1–5)	31 (29%)	7 (3–12)
Moderate/severe (≥6)	18 (17%)	11 (3–15)
Delusions (NMSS item 14), median (q1–q3)	0 (0–1)	
Not present	80 (75%)	
Mild (1–5)	14 (13%)	8 (3–13)
Moderate/severe (≥6)	13 (12%)	5 (0–13)
APPLIQue total score, median (q1–q3)		8 (3–12)
Partner		8 (4–14)
Daughter/son/sibling		2 (0–8)

PD, Parkinson’s disease; APPLIQue, the Alzheimer’s Patient Partners Life Impact Questionnaire (score range 0–25, higher = worse); q1–q3, first and third quartiles; HY, Hoehn and Yahr staging scale (score range I-V, higher = worse); S&E, Schwab & England Activities of Daily Living (ADL) scale (score range 0–100, higher = better); MMSE, Mini-mental state examination (score range 0–30, higher = better); UPDRS III, Unified PD Rating Scale, part III = motor examination (score range 0–108, higher = worse); NMSS, Non-Motor Symptoms Scale (0–360, higher = worse); GDS-30, Geriatric Depression Scale (score range 0–30, higher = worse). * 4 missing; ** 2 missing; *** 7 missing.

**Table 2 brainsci-12-00111-t002:** Simple linear regression analyses with APPLIQue as the dependent variable, *n* = 74.

Independent Variables	Unstandardized	Standardized	*p*-Value
	Coefficient β (95% CI)	Coefficient β	
Patient data			
Age (year)	−0.198 (−0.400 to 0.003)	−0.225	0.053
PD duration	−0.141 (−0.336 to 0.054)	−0.167	0.155
Patient gender (ref = male)	−3.935 (−6.744 to −1.125)	−0.313	**0.007**
Partner	4.017 (0.603 to 7.431)	0.266	**0.022**
Dwelling place (home vs. nursing home)	−1.252 (−4.195 to 1.691)	−0.099	0.399
Hoehn and Yahr stage	1.200 (−2.106 to 4.507)	0.085	0.472
ADL independency (S&E)	−0.022 (−0.126 to 0.082)	−0.05	0.674
Home health care (yes)	0.950 (−2.427 to 4.328)	0.066	0.577
Cognition, MMSE	−0.088 (−0.324 to 0.148)	−0.09	0.458
Cognitive impairment (dichotomized 0–23 vs. 24–30)	−4.271 (−7.135 to −1.407)	−0.337	**0.004**
Depressive symptoms (GDS-30)	−0.065 (−0.295 to 0.166)	−0.069	0.578
Depression (dichotomized 0–9 vs. 10–30)	−0.561 (−3.549 to 2.428)	−0.046	0.709
Motor function (UPDRS III total score)	0.039 (−0.053 to 0.132)	0.099	0.399
Non-motor symptomatology (NMSS total score)	0.036 (0.008 to 0.065)	0.292	**0.012**
Hallucinations (NMSS item 13)	0.206 (−0.209 to 0.621)	0.116	0.325
Hallucinations (dichotomized 0 vs. ≥ 1)	1.368 (−1.531 to 4.267)	0.11	0.350
Delusions (NMSS item 14)	0.089 (−0.546 to 0.723)	0.033	0.781
Delusions (dichotomized 0 vs. ≥ 1)	1.167 (−2.364 to 4.698)	0.077	0.512

CI, confidence interval; PD, Parkinson’s disease; APPLIQue, the Alzheimer’s Patient Partners Life Impact Questionnaire (score range 0–25, higher = worse); q1–q3, first and third quartiles; ref, reference category; HY, Hoehn and Yahr staging scale (score range I-V, higher = worse); ADL, activities of daily living; S&E, Schwab and England ADL scale (score range 0–100, higher = better); MMSE, Mini-mental state examination (score range 0–30, higher = better); NMSS, Non-Motor Symptoms Scale (0–360, higher = worse); GDS-30, Geriatric Depression Scale (score range 0–30, higher = worse); UPDRS III, Unified PD Rating Scale, part III = motor examination (score range 0–108, higher = worse). Bold *p*-values are statistically significant at *p* < 0.05.

**Table 3 brainsci-12-00111-t003:** Multivariable linear regression analyses with APPLIQue as the dependent variable, *n* = 70.

Independent Variables	Unstandardized	Standardized	*p*-Value
	Coefficient β (95% CI)	Coefficient β	
Patient data			
Cognitive impairment (MMSE dichotomized 0–23 vs. 24–30)	−4.139 (−6.880 to −1.397)	−0.327	**0.004**
Patient gender (ref = male)	−2.877 (−5.545 to −0.209)	−0.236	**0.035**
Age	−0.164 (−0.349 to 0.022)	−0.193	0.082

CI, confidence interval; PD, Parkinson’s disease; APPLIQue, the Alzheimer’s Patient Partners Life Impact Questionnaire (score range 0–25, higher = worse). MMSE, Mini-mental state examination (score range 0–30, higher = better). Ref = reference category. Bold *p*-values are statistically significant at *p* < 0.05. Adjusted R^2^ = 0.184. Independent variables entered in the multivariable linear regression model (backward method): age, gender, PD duration, partner, cognitive impairment (MMSE dichotomized at 23/24) and total NMS burden (NMSS total score).

## Data Availability

The data presented in this study are available on request from the corresponding author.

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
