# Peer review of "Caregiver Burden and Quality of Life in Late Stage Parkinson’s Disease"

_brainsci, 2022, doi:10.3390/brainsci12010111_

Round 1

Reviewer 1 Report

The manuscript presented novel data regarding caregiver burden of informal caregiver and quality of life in late stage PD patients. 74 informal caregiver from a cohort of 107 PD patients were assessed. Linear regression analyses showed better QoL associated with male gender, improved cognition and lower non-motor symptoms. The first two items were confirmed via multivariable regression analyses.

Comments:

1. General caregivers’ scores using caregiver burden inventory (CBI,  (Zarit et al. 1980).) and short form 36 health survey (SF-36 Ware and Sherbourne 1992) may be mentioned in the disussion as these were not used in this study. 

2. Line 98: n=62 (62%) were above the cutoff for depression. Is the percentage correct?

3. Table 1: Patient data (n=107) is shown, but in heading only n=74 (for informal caregiver) is mentioned. Number may be depicted clearer for reader in heading/first row of table.

4. Formatting of Table 3 may be improved (e.g. Unstandardized 
Coefficient β (95 % CI) in 1 row). P-Value column may be formatted accordingly. 

5. Two identical paragraphs in discussion part were detected: Lines 158-165 and Lines 180-186 are identical. Paragraph  should be used only at most relevant position. 

6. A Parkinson's disease caregiver burden questionnaire (PDCB) was recently developed for PD-specific measure of caregiver burden (Zhong et al. 2013). This may be mentioned in the limitations/discussion part. (ZhongM.EvansA.PeppardR. and VelakoulisD. (2013). Validity and reliability of the PDCB: a tool for the assessment of caregiver burden in Parkinson’s diseaseInternational Psychogeriatrics2514371441. doi: 10.1017/S1041610213000586.) This questionnaire was also previously validated in German and used for assessing caregiver burden in a German cohort. (Klietz, M., Schnur, T., Drexel, S., Lange, F., Tulke, A., Rippena, L., Paracka, L., Dressler, D., Höglinger, G. U., & Wegner, F. (2020). Association of Motor and Cognitive Symptoms with Health-Related Quality of Life and Caregiver Burden in a German Cohort of Advanced Parkinson's Disease Patients. Parkinson's disease2020, 5184084. https://doi.org/10.1155/2020/5184084)

7. Suggestion for additional references to improve discussion part:

7.1 Gender-specific caregiver burden was analyzed in German cohort previously and showed no difference. (Klietz M, von Eichel H, Schnur T, Staege S, Höglinger GU, Wegner F, Stiel S. One Year Trajectory of Caregiver Burden in Parkinson's Disease and Analysis of Gender-Specific Aspects. Brain Sci. 2021 Feb 26;11(3):295. doi: 103390/brainsci11030295.)

 7.2 Future interventions for caregiver burden may include mindfulness and psychological flexibility trainings to help PD caregivers as suggested previously. (Klietz M, Drexel SC, Schnur T, Lange F, Groh A, Paracka L, Greten S, Dressler D, Höglinger GU, Wegner F. Mindfulness and Psychological Flexibility are Inversely Associated with Caregiver Burden in Parkinson's Disease. Brain Sci. 2020 Feb 20;10(2):111. doi: 10.3390/brainsci10020111.)

Reviewer 2 Report

This paper explore an important and interesting problem-caregiver burden and informal caregiver QoL in late state PD. I have a few comments. 

Major comment: Multivariable linear regression analyses should enter key variables related to the caregiver burden including ADL, cognitive impairment, Motor function, etc to assess relation between caregiver burden (statistically significant or not) and caregiver quality of life while adjusting for other variables such as patient gender and age.

Minor comments

  1. Please check Table 2 and Table 3 to make sure the parameter estimates corresponding to the specific variable to avoid misunderstanding.
  2. For discussion, lines 158-165 in paragraph 3 and lines 180-186 in paragraph5 are repeated about cognitive function.
